# Impact of the COVID-19 Pandemic on the Prescribing Patterns of First-Line Antibiotics in English Primary Care: A Longitudinal Analysis of National Prescribing Dataset

**DOI:** 10.3390/antibiotics10050591

**Published:** 2021-05-17

**Authors:** Alisha Zubair Hussain, Vibhu Paudyal, Muhammad Abdul Hadi

**Affiliations:** School of Pharmacy, Institute of Clinical Sciences, University of Birmingham, Birmingham B15 2TT, UK; AXH995@student.bham.ac.uk (A.Z.H.); V.Paudyal@bham.ac.uk (V.P.)

**Keywords:** COVID-19, antibiotics, prescribing

## Abstract

The COVID-19 pandemic has impacted on public access to health services. This study aimed to investigate the impact of COVID-19 pandemic on commonly prescribed first-line antibiotics in English primary care. A secondary analysis of publicly available government data pertaining to primary care prescribing was conducted. A list of twenty first-line antibiotics used to treat common infections was developed following the National Institute of Clinical Excellence (NICE) guidelines. All primary care prescription and cost data pertaining to commonly prescribed first-line antibiotics in England between March and September of 2018–2020 were extracted and adjusted for inflation. Analysis suggests prescribing of antibiotics significantly reduced by 15.99% (*p* = 0.018) and 13.5% (*p* = 0.002) between March and September 2020 compared with same time period for 2018 and 2019, respectively. The most noticeable decrease in 2020 was noticed for prescribing for meningitis (−62.3%; *p* = 0.002) followed by respiratory tract infections (−39.13%; *p* = 0.035), in terms of indications. These results are suggestive of reduced transmission of infections in the community due to national lockdowns, social distancing and hygiene practices. In addition, the impact of reduced face-to-face consultations in general practices needs to be investigated as a potential reason for reduced prescribing. The pandemic also offers an opportunity to rationalize antibiotics use in the community.

## 1. Introduction

Since its first emergence in December 2019, the coronavirus 2019 (COVID-19) has caused over 3.1 million deaths and has affected approximately over 148 million people worldwide as of 28 April 2021 [1]. These figures continue to rise as the SARS-CoV-2 continues to mutate, making it more transmissible and, in some cases, deadlier. In the UK, the virus has affected over 4.4 million people and caused over 150,000 deaths [2]. COVID-19 has not only put healthcare systems globally under immense pressure but also changed how individuals live their lives, socialize and work due to different restrictions to prevent the transmission and limit the loss of lives. 

In the UK, the National Health Services (NHS) had to adapt to both the provision and nature of services in order to cope with the challenges of COVID-19. The NHS was provided with an additional £14 billion fund to help with the response to the coronavirus. In the UK, the first national lockdown began on the 23 March 2020 and continued till the 4 July 2020 [3]. Although the lockdown ended on the 4 July, local restrictions continued, including lockdowns in areas of high COVID-19 infection rates and limiting social gatherings to six people [3]. From 5 March 2020, NHS England advised general practices to close their doors to face-to-face consultations where such consultations could be undertaken virtually and encouraged remote consultations [4,5]. Subsequently, a 51.5% decrease in the number of face-to-face appointments within the primary care settings was noted between April and August 2020 compared to the same period in 2019. However, the number of telephone appointments increased over 2.5 folds during the same period [6]. These drastic changes in the delivery of services in primary care, such as general practices and community pharmacies, have raised concerns with regards to timely equitable access to these services [5,7]. 

Antibiotics consumption within the primary care setting makes up a large proportion of all antimicrobial utilization in the UK, equating to 72% of all antibiotic prescribing in primary care [8]. The appropriate use of antibiotics in bacterial infection during COVID-19 has bought about several challenges. Challenges associated with remote consultations, such as examining the signs and symptoms of infection, as well as limited access to culture and sensitivity test results, may make rationale prescribing decisions difficult, especially for practitioners with limited exposure to the concept of remote consultation [9]. Furthermore, the overlapping symptoms of COVID-19 with respiratory tract infection and the lack of availability of vast COVID-19 testing in the beginning of the pandemic might have led to an initial increase in prescribing antibiotics for respiratory tract infections. On the other hand, the national lockdown, school/university closures and restricted social gatherings could have reduced the prevalence of transmissible infections. Additionally, behavioural changes, such as frequent hand washing, wearing masks in social places and social distancing, might have also reduced the prevalence of infectious diseases. However, there is currently limited empirical evidence to document the actual impact of COVID-19-related restrictions on prescribing of antibiotics. An observational study from the Netherlands reported a reduction in the presentation of bacterial infections and antibiotic prescribing in primary care [10]. This study was designed to further build on existing evidence and aimed to assess the impact of COVID-19-related restrictions on antibiotic prescribing patterns and costs in English primary care.

## 2. Results

The results presented below are based on 7-month antibiotic prescribing data (March–September) for the years 2018, 2019 and 2020 in English primary care.

### 2.1. Prescribing of First-Line Antibiotics between 2018 and 2020

The overall number of first-line antibiotics prescribed in England for the treatment of common infections decreased by 15.99% (*p* = 0.018) and 13.5% (*p* = 0.002) between March and September in 2020 compared to the corresponding periods in 2018 and 2019, respectively (Table 1 and Figure 1). Amoxicillin was the most commonly prescribed antibiotic during 2018, 2019 and 2020. In 2020 (March–September), prescribing for amoxicillin decreased by 32.7% (*p* = 0.035) and 43.5% (*p* = 0.035) compared to 2018 and 2019, respectively. Although the prescribing volume was very low, the highest decrease in prescribing in 2020 was noticed for benzylpenicillin: 69.9% (*p* = 0.002) in 2018 and 62.3% (*p* = 0.002) in 2019 (Table 1). Benzylpenicillin in primary care is solely used for the treatment of meningitis, as stated in the NICE guidelines, and prescriptions decreased from 373 items in 2018 (between March and September) to 112 items in 2020 (between March and September).

### 2.2. Prescribing Patterns of First-Line Antibiotics by Pharmacological Drug Classification and Anatomical Infection Sites

Overall, antibiotic prescribing for all anatomical infection sites decreased in the months between March and September 2020 compared with same corresponding periods of 2018 and 2019. The most common condition for which antibiotics were prescribed across the three years was skin and soft tissue (28.61 million items prescribed) followed by lower respiratory tract infection (19.08 million items prescribed) (Table 2). For skin and soft tissue infections, the number of items prescribed decreased by 29.6% from 10.7 million items in 2018 (between March and September) to 7.5 million items in 2020 (*p* = 0.025) (Table 2). The biggest decrease in prescribing antibiotics was noticed for upper respiratory tract infections where the number of items prescribed in 2020 reduced by 42.13% (*p* = 0.025) and 39.13% (*p* = 0.035) compared with antibiotics prescribing in 2018 and 2019, respectively. Similarly, there was a significant reduction in the prescribing for dental infections in 2020 compared with 2018 (−40.95%; *p* = 0.025) and 2019 (−38.02%; *p*= 0.035).

In terms of pharmacological classification of the drugs, prescribing across all drug classes was reduced significantly during 2020 compared to 2018 and 2019 (All *p* < 0.05) (Table 3). Penicillins were the most commonly prescribed drug class across three years of the study. The prescribing accounted for 42.69%, 47.93% and 38.22% in 2018, 2019 and 2020, respectively. In 2020, the prescribing of penicillins decreased from 8.41 million items in 2018 to 6.47 million (−23.09%; *p* = 0.018). However, prescribing for cephalosporins increased from 0.43 million and 0.42 million in 2018 (*p* = 0.085) and 2019 (*p* = 0.025), respectively, to 0.45 million in 2020. 

### 2.3. Costs Associated with Prescribing of First-Line Antibiotics between 2018 and 2020

Although, there was some variation in the cost trends of individual drugs, the overall cost associated with first-line antibiotics prescribing decreased by 4.42% from £62,083,604.47 in 2018 (between March and September) to £59,342,464.69 in 2020 (Figure 2). Similarly, in comparison to 2019 (between March and September), a 5.18% reduction in the costs of antibiotics was noticed in 2020 (Table 4). In 2020, the highest reduction in cost was noticed for phenoxymethylpenicillin compared to 2018 (−48.21%; *p* = 0.006) and 2019 (28.90%; *p* = 0.018). Compared to 2019, the costs associated with prescribing of ciprofloxacin (2.50%; *p* = 0.620), ofloxacin (20.61%; *p* = 0.003), doxycycline (74.61; *p* = 0.002), clarithromycin (6.74; *p* = 0.277), clindamycin (6.45; *p* = 0.003), chloramphenicol (6.19%; *p* = 0.001) and co-amoxiclav (9.03%; *p* = 0.064) increased in 2020. 

## 3. Discussion

The overall aim of the study was to investigate the impact of restrictions related to COVID-19 on the prescribing of first-line antibiotics in English primary care. We found that the prescribing of selected first-line antibiotics in English primary care was significantly reduced in the months corresponding to the national lockdown and other local restrictions in 2020 compared to 2018 and 2020. Our findings are in line with the findings of other international studies. Studies from the Netherlands [10] and the USA [11,12] have reported a decrease in presentation of infections and antibiotic prescribing in 2020 compared to previous year(s). However, the Dutch study [10] reported the highest decline in prescriptions for gastrointestinal infection, in contrast with the findings of our study that reported the highest decline for upper respiratory tract prescriptions. Data from the USA reported the highest reduction in antibiotic prescribing in the outpatient setting in April (–39%) and May (–42%) 2020 compared to antibiotic prescribing in 2017–2019 [12]. However, the authors noticed an increase in consumption of azithromycin in February to March 2020. This increase in the consumption of azithromycin can be attributed to earlier evidence to support its use with hydroxychloroquine in the management of COVID-19 [13]. 

A number of potential factors, including the changes in the delivery of primary care services, reduction in the incidence to bacterial infections due to behavioural changes and lockdown, and fear and anxiety of attending face-to-face appointments can potentially explain the reduction in prescribing of antibiotics in England. Perhaps the simplest explanation for this reduction in antibiotic prescribing is the reduction in face-to-face consultations in 2020 compared to the preceding years [6]. As mentioned earlier, face-to-face appointments in English primary care decreased by over 50% between April and August 2020 compared to the same time period in 2019 [6]. Although the face-to-face consultations were largely replaced by remote consultations, patients with hearing loss, language barriers and those less able to use digital technology were disadvantaged in accessing primary care services [14]. Similarly, patients who are less verbally articulate were difficult to diagnose during a remote consultation, as a face-to-face consultation provides an opportunity for healthcare professionals to pick up on visual cues [14]. This may have hindered the ability to diagnose acute infection and hence led to the reduction in prescribing of first-line antibiotics. Furthermore, overall, the number of consultations in primary care has been estimated to drop to less than three consultations per year from four consultations per year per person [15]. This may have meant that fewer patients were seeking help for acute infections and subsequently did not receive antibiotics for common infections. A reason for this avoidance could have been due to the fear of the coronavirus infection. Furthermore, some patients avoided seeking help as they felt they would be creating a huge workload on the NHS due to the pandemic [16]. 

The decrease in first-line antibiotics prescribed may be reflective of the changes in public behaviour and activities during the pandemic as a result of the national lockdown and public health campaign to limit the spread of COVID-19. These strict measures included the closure of all non-essential shops, restaurants and bars, schools and universities and a ban on all social gatherings. Furthermore, public health campaigning on hand hygiene, face masks and sanitation to prevent the spread of the coronavirus also might have contributed to prevent the spread of bacterial infections in addition to COVID-19 [17]. Respiratory tract infections, which are transmitted through airborne droplets from coughing and sneezing as well as touching the nose or mouth by exposed objects, may have been reduced for these reasons. Hence, a reduced incidence of respiratory tract infections might have resulted in reduced prescribing of first-line antibiotics, including amoxicillin, phenoxymethylpenicillin, doxycycline and clarithromycin reported in this study. 

A relatively small reduction was found in prescribing of first-line antibiotics for urinary tract infection and genital tract infections. A potential reason for this finding is in the differences in transmission of these infections compared with respiratory tract infections. Urinary tract infections are mostly caused by *E. coli*, which is spread from the gut and relocates in the urinary tract [17]. Thus, measures taken to prevent the spread of the coronavirus, including the ban on social gatherings and wearing face masks, would not have impacted the number of infections. 

Prior to the pandemic, it is very well documented in the literature that antibiotics are often overprescribed for non-bacterial infections, such as influenza [18,19,20,21,22,23]. In addition to overprescribing, in other countries, although not a problem in the UK, over-the-counter sales of antibiotics without a prescription is also an important factor contributing to overuse of antibiotics, leading to antimicrobial resistance [24,25]. This overprescribing of antibiotics is one of the most significant drivers of antimicrobial resistance. Given that more than two thirds of antibiotics are consumed in primary care in England, this decrease in antibiotics prescribing due to COVID-19 has provided an opportunity to healthcare professionals to reflect on the prescribing of antibiotics [18]. 

### Strengths and Limitations

The study only included prescribing data of selected first-line antibiotics used within English primary care, and hence does not include antibiotics prescribed in hospital settings nor private prescriptions. Although this investigation included a wide scope of infections, some of the antibiotics were overlapping in multiple infections. The prescribing data provided by OpenPrescribing.net does not provide the indication of the use of the antibiotics included in this study. The outcomes for treatment were not included in this study. However, this was beyond the scope of the project. 

## 4. Materials and Methods

### 4.1. Study Design and Population

Secondary analysis on the prescribing and dispensing of 20 first-line antibiotics within the NHS England primary care setting, between March and September of 2018–2020, was undertaken as part of this study. The same months were chosen across three years to limit the impact of seasonal variations on the prescribing of antibiotics. This list of first-line antibiotics was obtained from the National Institute of Clinical Excellence (NICE) summary of antimicrobial prescribing guidance—managing common infections—March 2020 [16]. The National Institute of Clinical Excellence (NICE) guidelines’ summary of antimicrobial prescribing guidance—managing common infections—include treatments for upper respiratory tract infections, lower respiratory tract infections, urinary tract infection, meningitis, gastrointestinal tract infections, genital tract infections, skin and soft tissue infections, eye infections and dental infection in primary care [26]. This guideline included 20 antibiotics used for first-line infections, including amoxicillin, phenoxymethylpenicillin, doxycycline, clarithromycin, co-amoxiclav, nitrofurantoin, trimethoprim, ciprofloxacin, ofloxacin, cefalexin, benzylpenicillin, metronidazole, metronidazole cream, vancomycin, clindamycin, ceftriaxone, fusidic acid, mupirocin, flucloxacillin and chloramphenicol [26]. The first-line antibiotics were included in the study to give an estimate of the prevalence of the use of antibiotics in the treatment of bacterial infection. The NICE guideline recommends a further 12 antibiotics that are used as second- and third-line therapy, which were excluded in the study [26]. 

Data on antibiotic prescribing and costs were obtained from OpeningPrescribing.net. OpenPrescribing.net is an Evidence-Based Medicine (EBM) DataLab at the University of Oxford [27]. The EBM DataLab publishes anonymised data provided by the NHS Business and Service Authority on the total number of items of each drug prescribed and the total spending across all practices in England. A number of studies have used the same dataset to study the prescribing patterns of various classes of drugs [28,29].

### 4.2. Data Extraction and Analysis

Prescribing data from 2018–2020 (focusing on the months March to September) for antibiotics were extracted from OpenPrescribing.net into Microsoft excel. The data were adjusted for population estimates for each year in England using the mid-year 2019 population estimates published by the ONS. The cost of each antibiotic prescription was adjusted for inflation using the ONS Consumer Price Index [30]. All data were extracted, independently checked for accuracies, and analysed using Microsoft Excel and IBM SPSS Statistics (version 27). Mann–Whitney U-tests (the non-parametric equivalent of Student’s *t*-test) were used to examine the differences in the prescribing patterns and costs of antibiotics between 2018 and 2020, and between 2019 and 2020. A two-tailed *p*-value of less than 0.05 was considered statistically significant.

## 5. Conclusions

COVID-19-related restrictions significantly reduced prescribing of all first-line antibiotics in English primary care in 2020 compared with preceding years. Further work is needed to establish the extent of the contribution made by reduced access to services, inability of prescribers to make an informed assessment and patient reluctance to use health services in reducing the prescribing of antibiotics.

## Figures and Tables

**Figure 1 antibiotics-10-00591-f001:**
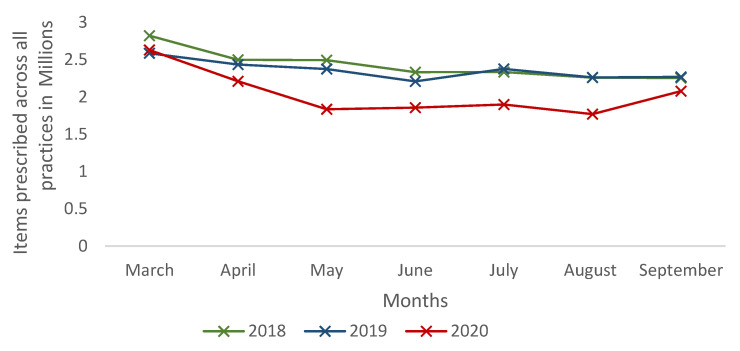
Number of first-line antibiotics prescribed in English primary care.

**Figure 2 antibiotics-10-00591-f002:**
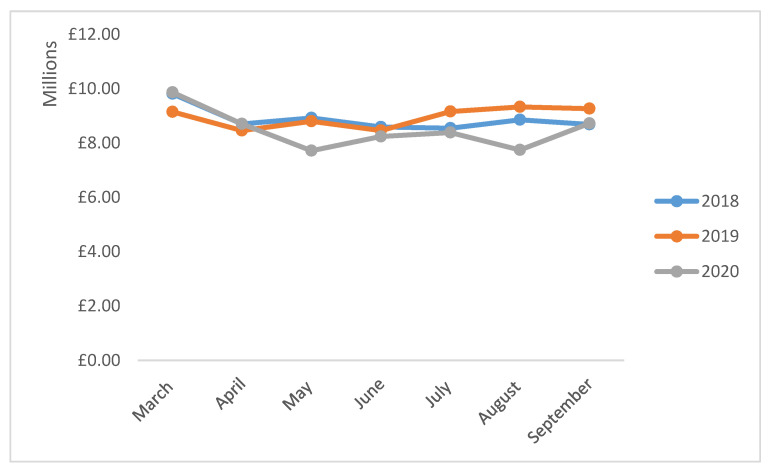
Costs associated with prescribing of first-line antibiotics in English primary care.

**Table 1 antibiotics-10-00591-t001:** Number of first-line antibiotics prescribed in English primary care.

Antibiotic	Items Prescribed (March–September)			% Difference 2020 and 2018 (*p*-Value) *	% Difference 2020 and 2019 (*p*-Value) *
	2018	2019	2020		
Amoxicillin	3,788,146	3,656,355	2,548,155	−32.7 (0.035)	−43.5 (0.035)
Phenoxymethylpenicillin	1,345,567	1,224,614	938,718	−30.2 (0.048)	−23.35 (0.025)
Doxycycline	1,418,699	1,552,700	1,512,059	6.6 (0.949)	−2.6 (0.225)
Clarithromycin	1,150,884	1,114,988	914,060	−20.6 (0.048)	−18.0 (0.064)
Co-amoxiclav	781,027	733,962	730,055	−6.5 (0.085)	−1.0 (0.565)
Nitrofurantoin	2,296,145	2,436,156	2,418,624	5.3 (0.048)	−0.7 (0.749)
Trimethoprim	1,131,986	952,513	892,998	−21.1 (0.002)	−6.3 (0.009)
Ciprofloxacin	333,774	280,746	258,309	−22.6 (0.002)	−7.9 (0.013)
Ofloxacin	20,222	19,525	23,413	15.8 (0.004)	19.9 (0.002)
Cefalexin	428,462	422,528	445,779	4.0 (0.0085)	5.5 (0.025)
Benzylpenicillin	373	297	112	−69.9 (0.002)	−62.3 (0.002)
Metronidazole (Oral)	337,999	331,865	304,898	−9.8 (0.002)	−8.1 (0.002)
Vancomycin	2710	2417	3152	16.3 (0.018)	30.4 (0.013)
Metronidazole (Topical)	17,028	19,821	21,406	25.7 (0.002)	8.0 (0.064)
Clindamycin (Topical)	18,676	17,213	15,139	−18.9 (0.002)	−12.1 (0.002)
Fusidic acid	637,990	627,026	473,193	−25.8 (0.002)	−24.5 (0.002)
Mupirocin	38,692	38,398	31,038	−19.8 (0.003)	−19.8 (0.002)
Flucloxacillin	2,502,571	2,388,848	2,256,687	−9.8 (0.110)	−5.5 (0.655)
Ceftriaxone	828	616	570	−31.2 (0.005)	−7.5 (0.749)
Chloramphenicol	735,945	683,392	482,716	−34.4 (0.004)	−29.4 (0.002)
All antibiotics	16,987,724	16,503,980	14,271,081	−15.99 (0.018)	−13.5 (0.002)

* *p*-Values calculated using Mann–Whitney U-tests.

**Table 2 antibiotics-10-00591-t002:** Number of antibiotics prescribed (in millions) as per anatomical infection sites.

	Number of Items Prescribed between March and September in Millions *			% Difference 2020–2018 (*p*-Value) *	% Difference 2020–2019 (*p*-Value) *
	2018	2019	2020		
Upper respiratory tract	5.13	4.88	2.97	−42.13 (0.025)	−39.13 (0.035)
Lower respiratory tract	7.13	7.05	4.90	−31.40 (0.064)	−30.62 (0.085)
Urinary tract infection	4.99	4.84	4.07	−18.52 (0.002)	−16.06 (0.002)
Meningitis	0.0004	0.0003	0.00009	−69.9 (0.002)	−62.3 (0.002)
Gastro-intestinal tract	6.07	5.84	3.85	−36.42 (0.035)	−34.01 (0.048)
Genital tract	2.14	2.22	1.84	−14.33 (0.482)	−17.23 (0.18)
Skin and soft tissue	10.67	10.44	7.50	−29.59 (0.025)	−28.17 (0.035)
Eye infection	0.74	0.69	0.42	−42.63 (0.004)	−38.22 (0.002)
Dental infection	5.47	5.21	3.23	−40.95 (0.025)	−38.02 (0.035)

* *p*-Values calculated using Mann–Whitney U-tests.

**Table 3 antibiotics-10-00591-t003:** Number of antibiotics prescribed (in millions) as per pharmacological class of antibiotics between March and September of the respective years.

	Number of Items Prescribed in Millions			% Difference 2018–2020 (*p*-Value)*	% Difference 2019–2020 (*p*-Value) *
	2018	2019	2020		
Penicillin	8.41	8.00	6.47	−23.09 (0.018)	−19.12 (0.025)
Cephalosporins	0.43	0.42	0.45	3.98 (0.085)	5.49 (0.025)
Macrolides	1.15	1.11	0.91	−20.58 (0.048)	−18.02 (0.064)
Quinolones	0.35	0.30	0.28	−20.42 (0.002)	−6.18 (0.0035)
Tetracyclines	1.42	1.55	1.51	6.58 (0.949)	−2.62 (0.225)
Others	5.41	5.30	4.80	−11.19 (0.002)	−9.43 (0.002)

* *p*-Values calculated using Mann–Whitney U-tests.

**Table 4 antibiotics-10-00591-t004:** Costs (in millions) of the antibiotics prescribed between March and September in 2018–2020.

Antibiotic	Costs in Millions			% Difference 2020–2018 (*p*-Value) *	% Difference2020–2019 (*p*-Value) *
	2018	2019	2020		
Amoxicillin	4.12	4.49	4.26	3.18 (0.482)	−5.22 (0.277)
Phenoxymethylpenicillin	7.46	5.43	3.86	−48.21 (0.006)	−28.90 (0.018)
Co-amoxiclav	1.82	1.87	2.04	11.90 (0.013)	9.03 (0.064)
Benzylpenicillin	0.004	0.002	0.002	−57.59 (0.025)	−17.33 (0.085)
Flucloxacillin	11.24	11.29	8.13	−27.70 (0.002)	−28.05 (0.002)
Cefalexin	0.73	0.78	1.06	45.78 (0.002)	36.27 (0.002)
Ceftriaxone	0.08	0.06	0.05	−36.85 (0.006)	−15.70 (0.085)
Ciprofloxacin	0.90	0.84	0.86	−4.90 (0.482)	2.50 (0.620)
Ofloxacin	0.78	0.72	0.87	12.58 (0.004)	20.61 (0.003)
Doxycycline	1.75	2.38	4.16	137.37 (0.002)	74.61 (0.002)
Clarithromycin	2.81	2.81	3.00	6.54 (0.482)	6.74 (0.277)
Clindamycin	1.12	0.82	0.87	−22.06 (0.002)	6.45 (0.003)
Nitrofurantoin	20.91	20.96	20.81	−0.489 (0.949)	−0.75 (0.902)
Trimethoprim	1.59	1.78	1.82	14.62 (0.064)	2.70 (1.000)
Metronidazole oral	1.68	2.08	4.25	−5.74 (0.655)	−23.87 (0.001)
Metronidazole ointment	2.11	2.19	3.86	−16.86 (0.655)	−19.98 (0.001)
Vancomycin	0.57	0.50	2.04	−13.64 (0.225)	−1.48 (0.710)
Fusidic acid	1.33	1.62	0.002	33.24 (0.002)	8.93 (0.142)
Mupirocin	0.24	0.23	8.12	−18.78 (0.003)	−18.20 (0.002)
Chloramphenicol	1.24	1.79	1.07	52.88 (0.003)	6.19 (0.001)
All antibiotics	62.08	62.58	59.34	−4.42% (0.180)	−5.18 (0.097)

* *p*-Values calculated using Mann–Whitney U-tests.

## Data Availability

The data presented in the paper can be found at https://openprescribing.net/ (accessed on 28 April 2021).

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
