# Peer review of "Impact of the COVID-19 Pandemic on the Prescribing Patterns of First-Line Antibiotics in English Primary Care: A Longitudinal Analysis of National Prescribing Dataset"

_antibiotics, 2021, doi:10.3390/antibiotics10050591_

Round 1

Reviewer 1 Report

Well written and easy to follow. This reviewer has no major comments. As minor comments, I suggest to highlight effect sizes (% differences in this case) more than p-values (when comparing large groups there could be a statistically significant difference even for very small % differences the may be not be meaningful/relevant). I also suggest to discuss more in-depth why there were less prescription despite the important and inevitable increase in phone visits (one may argue that without visiting the patient there may be more diagnostic uncertainty and thus increased willingness to prescribe an antibiotic). Finally, I agree many antibiotics are prescribed uselessly for viral infections such as influenza. However, also COVID-19 is a viral respiratory disease and many were unfortunately affected during the study period. The authors may want to discuss why there was a decrease in antibiotic prescriptions despite the high prevalence of COVID-19, which may resemble influenza in mild forms.

Author Response

Reviewer's comment: Well written and easy to follow. This reviewer has no major comments. As minor comments, I suggest to highlight effect sizes (% differences in this case) more than p-values (when comparing large groups there could be a statistically significant difference even for very small % differences the may be not be meaningful/relevant). I also suggest to discuss more in-depth why there were less prescription despite the important and inevitable increase in phone visits (one may argue that without visiting the patient there may be more diagnostic uncertainty and thus increased willingness to prescribe an antibiotic). Finally, I agree many antibiotics are prescribed uselessly for viral infections such as influenza. However, also COVID-19 is a viral respiratory disease and many were unfortunately affected during the study period. The authors may want to discuss why there was a decrease in antibiotic prescriptions despite the high prevalence of COVID-19, which may resemble influenza in mild forms.

Authors' response: Thank you very much for reviewing our paper and providing constructive feedback. We agree that the percentage change is more important than P-values. Therefore, in all tables we have provided both percentage changes and p-values. Also in the text we have provided percentage changes in antibiotic prescribing where applicable.  

We have already discussed low prescribing despite remote consultation in the discussion section. Please refer to lines 163-170 and furthermore the same issue is discussed in lines 179- 190

Reviewer 2 Report

Alisha Zubair Hussain, et al, presented this paper entitled "Impact of COVID-19 Pandemic on Prescribing Patterns of First line Antibiotics in English Primary care: A Secondary Analysis". It is a very interesting paper dealing with an up-to-date issue. It confirms the quite unexpected data of reduced trend of antibiotic prescription during the cover-19 pandemic. I think the paper is well-written and the methodology adequate. 

I advice authors to add as reference and to comment that paper from King LM, et al. "Trends in US Outpatient Antibiotic Prescriptions During the Coronavirus Disease 2019 Pandemic. CID 2021", as it provides similar data.

Finally two minor the error are present along the text in lines 103 and 111.

Author Response

Reviewer's comment: Alisha Zubair Hussain, et al, presented this paper entitled "Impact of COVID-19 Pandemic on Prescribing Patterns of First line Antibiotics in English Primary care: A Secondary Analysis". It is a very interesting paper dealing with an up-to-date issue. It confirms the quite unexpected data of reduced trend of antibiotic prescription during the cover-19 pandemic. I think the paper is well-written and the methodology adequate. 

I advice authors to add as reference and to comment that paper from King LM, et al. "Trends in US Outpatient Antibiotic Prescriptions During the Coronavirus Disease 2019 Pandemic. CID 2021", as it provides similar data.

Finally two minor the error are present along the text in lines 103 and 111.

Authors reply: Thank you very much for considering our paper and providing constructive feedback. We have now incorporated the above mentioned reference and discussed their findings in relation to our study findings. Minor typo issues in line 103 and 111 have also been fixed.